# Alternative Splicing in Angiogenesis

**DOI:** 10.3390/ijms20092067

**Published:** 2019-04-26

**Authors:** Elizabeth Bowler, Sebastian Oltean

**Affiliations:** Institute of Biomedical and Clinical Sciences, Medical School, College of Medicine and Health, University of Exeter, Exeter EX4 4PY, UK; e.bowler@exeter.ac.uk

**Keywords:** alternative splicing, angiogenesis, VEGF, VEGFR, NRP, FGFR, vasohibin, HIF-1, angiopoietin

## Abstract

Alternative splicing of pre-mRNA allows the generation of multiple splice isoforms from a given gene, which can have distinct functions. In fact, splice isoforms can have opposing functions and there are many instances whereby a splice isoform acts as an inhibitor of canonical isoform function, thereby adding an additional layer of regulation to important processes. Angiogenesis is an important process that is governed by alternative splicing mechanisms. This review focuses on the alternative spliced isoforms of key genes that are involved in the angiogenesis process; *VEGF-A, VEGFR1, VEGFR2, NRP-1, FGFRs, Vasohibin-1, Vasohibin-2, HIF-1α, Angiopoietin-1* and *Angiopoietin-2*.

## 1. Alternative Splicing

The human genome consists of an estimated 20,000 protein-coding genes [1], which can give rise to a much greater number of proteins. Alternative splicing is one of the main processes that accounts for the increased protein to gene ratio. In fact, alternative splicing is reported to be responsible for 40% of protein modifications [2]. It involves the formation of multiple mRNA products from a single pre-mRNA, and many of these mRNA products are then translated into proteins that may have distinct structures or functions [3]. This important process therefore increases the coding capacity of a given gene, and as it has been reported that more than 95% of pre-mRNAs are alternatively spliced in mammals [4], research into alternative splicing has gathered momentum over the past two decades. Alternative splicing is heavily regulated. However, like most biological processes, alternative splicing is open to faults, which can result in disease. It has been suggested that around 33% of mutations that cause disease affect splicing [5]. 

### 1.1. The Splicing Reaction

Splicing of pre-mRNAs involves the excision of introns and joining together of exons, which forms the mature mRNA transcript that is then translated into protein [6]. This is achieved by the spliceosome, which is a macromolecule structure comprised of five small nuclear ribonucleoprotein particles (snRNPs) and over 100 proteins [7]. 

The splicing reaction occurs in multiple stages, as outlined in Figure 1. Briefly, the *CAG GURAGU* consensus sequence at the 5′ of the intron is bound by the U1 snRNP, whilst SF1 binds to the branch point sequence *YNYURAC*, and the 35kDa U2AF subunit (U2AF35) binds to the 3′ splice site via an *AG* dinucleotide. In addition, the downstream polypyrimidine sequence of the branch point is bound by the U2AF65 subunit. The U2 snRNP displaces the SF1 protein at the branch point sequence. Following this, the U5.U4/U6 tri-snRNP is recruited to the U1 snRNP, and the NineTeen Complex (NTC) connects to the spliceosome. After conformational and compositional rearrangements, the catalytic function of the spliceosome is activated, and a transesterification reaction occurs. This releases the U1 and U4 snRNPs and forms an intron lariat at the 3′ part of the exon. A further transesterification reaction at the 3′ splice site releases the 3′ exon, which leads to exon ligation and excision of the lariat containing U4, U5 and U6 snRNPs. The spliced mature mRNA transcript can then be translated into protein [7,8].

### 1.2. Regulation of Alternative Splicing

Alternative splicing arises when exons or introns are either included or excluded from a mature mRNA transcript. The major alternative splicing patterns (exon skipping, intron retention, mutually exclusive exons and alternative 3′ or 5′ splice sites) are illustrated in Figure 2, which forms through the recognition of short *cis*-acting splicing regulatory elements (SREs) by *trans*-acting splice factors or RNA binding proteins (RBPs). SREs are categorised depending on where in the pre-mRNA the RNA binding protein is binding (exon or intron), and whether they enhance or inhibit splicing. This forms four subgroups of SREs; exonic splicing enhancers (ESEs), intronic splicing enhancers (ISEs), exonic splicing silencers (ESSs) and intronic splicing silencers (ISSs). 

RBPs recognise particular SREs and determine which sections of pre-mRNA are included in the mature transcript. Therefore, the regulation of alternative splicing can be influenced by the amount and activity of RBPs, the two main studied groups of which are serine/arginine rich (SR) proteins (12 discovered to date [9]) and heterogenous nuclear ribonucleoproteins (hnRNPs) (23 discovered to date [10]). Originally, it was thought that SR proteins bind to ESEs and promote exon inclusion, whereas spliceosome access to the polypyrimidine tract is blocked by hnRNPs, thereby promoting exon exclusion [11]. For example, hnRNP H/F has been shown to bind to exon 18b regulatory sequences of T-cell factor-3, which represses the inclusion of exon 18b and results in an increase in the E12 isoform that contains exon 18a instead of exon 18b [12]. However, instances whereby SR proteins promote exon exclusion and hnRNP proteins promote exon inclusion have since been reported [13,14,15,16]. In addition, it has been reported that SR proteins and hnRNPs are antagonists of each other when binding to SREs [17]. Therefore, changes in the ratio of SR protein to hnRNPs in the nucleus can alter splicing. 

The SR proteins SRSF1, SRSF2 and SRSF6 are involved in the regulation of the spliceosome itself. They promote the binding of the U1 and U2 snRNPs to the 5′ and 3′ splice sites, respectively [18]. Therefore, the binding of SR proteins to SREs is highly regulated through phosphorylation activation by SR protein kinases, such as CDC-like-kinase 1 (CLK1) and SRSF-protein-kinase 1 (SRPK1). For instance, an established pathway for SRSF1-mediated splicing involves phosphorylation of SRSF1 by SRPK1, which facilitates the binding of SRSF1 to a transportin protein (Tr-SR) and results in the nuclear import of SRSF1 [19]. Once in the nucleus, SRSF1 is stored as nuclear speckles, and subsequent hyper-phosphorylation by CLK1 releases SRSF1 from nuclear speckle storage and allows SRSF1 to bind to SREs on pre-mRNA for the splicing reaction [20]. In addition to the major alternative splicing patterns usually documented, two further alternative splicing mechanisms are also depicted in Figure 2; alternative polyadenylation and alternative promoters. Multiple polyadenylation sites can exist throughout a pre-mRNA sequence, which can be cut prematurely to the canonical isoform, and a polyadenylation tail is added. A recent model of *VEGFR-1* mRNA alternative polyadenylation regulation has been proposed whereby demethylated hnRNP D is unable to bind to ARE in intron 13 [21]. This results in the premature cutting of the pre-mRNA by cleavage factors and the addition of the poly-A tail to the truncated pre-mRNA, which produces the soluble *VEGFR-1* (*sVEGFR-1*) isoform [21]. However, when hnRNP D is methylated, it binds to ARE in intron 13 and causes the splicing out of the intron, which produces the membrane-bound canonical isoform of *VEGFR-1* [21]. 

## 2. Angiogenesis

Angiogenesis is an important process that is defined as the formation of new blood vessels from pre-existing vasculature. It is vital during embryonic development, wound healing, hair growth, and reproduction [22]. However, dysregulation of angiogenesis is associated with a range of pathologies in adults, such as cancer, diabetic retinopathy, rheumatoid arthritis and endometriosis [23,24]. This has led to the study of complex signalling pathways that regulate angiogenesis. 

### Angiogenic Vessel Formation

The angiogenesis process is activated by pro-angiogenic signals, such as vascular endothelial growth factor-A (VEGF-A), angiopoietin-2 (ANG-2) and fibroblast growth factor (FGF) [25]. Ang-2 and matrix metalloproteinases (MMPs) mediate the detachment of pericytes from the vessel wall. This causes the loosening of junctions that are between the endothelial cell monolayer and the nascent vessel dilates. VEGF-A, which is the most extensively investigated member of the VEGF family, increases endothelial cell layer permeability which forces plasma proteins out of the blood vessels and into the surrounding tissue, where they form an extracellular matrix scaffold for the attachment of endothelial cells. The formation of the vascular sprout is guided by the tip cell (stimulated by VEGF-A receptors, neuropilins, Delta-like 4 (DLL4) and JAGGED1 (JAG1), and the elongation of the stalk is facilitated by neighbouring stalk cells (stimulated by NOTCH, NOTCH regulated Ankyrin repeat protein (NRARP), wingless/integrateds (WNT)s, placental growth factor (PIGF) and fibroblast growth factors (FGFs)) [26]. Recently, the macrophage stimulating-1/Forkhead box protein O1 (MST1/FOXO1) cascade was identified to direct the migration of tip cells towards hypoxic regions [27]. The vascular lumen is then established and this is mediated by a variety of factors including: vascular endothelial cadherin (VE-cadherin), CD34, sialomucins, VEGF-A and hedgehog ligands. Following this, the cells become covered in pericytes, which is signalled by the expression of factors, such as platelet-derived growth factor-β (PDGF-B), angiopoietin-1 (ANG-1), transforming growth factor-β (TGF-β), ephrin-B2 and NOTCH. The basement membrane is reformed through the inhibition of MMPs by a tissue inhibitor of metalloproteinases (TIMPs) and plasminogen activator inhibitor-1 (PaI-1), which results in the maturation of the vessel [26]. The important process of angiogenesis is highly regulated at many levels, including transcriptional [28,29], post-transcriptional [30], microRNAs [30,31], allosteric enhancement [32], and as discussed in this review, alternative splicing. Many genes involved in angiogenesis have been shown to be alternatively spliced, and there are many instances whereby isoforms negatively regulate the canonical isoform (Figure 3). This review will discuss current known splicing events that occur in genes associated with angiogenesis and the functions of their splice isoforms (if known) as shown in Table 1. The major splicing events are shown in Figure 4.

## 3. Splicing in Angiogenesis

### 3.1. Vascular Endothelial Growth Factor-A (VEGF-A)

The binding of VEGF-A to VEGF receptor 2 (VEGFR2) is considered to be the main stimulatory signal of angiogenesis in vivo [60]. Studies have shown that VEGF-A is secreted by many different cell types, including: endothelial cells, fibroblasts, smooth muscle cells, platelets, neutrophils and macrophages [34]. VEGF-A is produced by 60% of tumours [61] and has therefore been extensively investigated for anti-cancer therapeutics.

The *VEGF-A* mRNA contains eight exons, the splicing of which gives rise to a variety of isoforms. The six major isoforms of *VEGF-A* are: *VEGF-A_111_, VEGF-A_121_, VEGF_145_, VEGF-A_165_, VEGF-A_189_* and *VEGF-A_206_* (where the subscripted numbers denoted the number of amino acids present in the isoform) [34]. Additionally, another less commonly expressed isoform has also been reported, known as *VEGF-A_183_* [62]. The bioavailability of the isoforms is governed by a mRNA sequence that spans exons 6a and 7 determines the binding capacity of the isoforms to heparin and heparin sulphate proteoglycans that are located on the cell surface and extracellular matrix [34,63]. VEGF-A_145_, VEGF-A_189_ and VEGF-A_206_ strongly bind to both the cell surface and extracellular matrix, whereas VEGF-A_111_ and VEGF-A_121,_ which both lack exons 6 and 7, are readily diffusible [35,36,37]. VEGF-A_165_ is moderately diffusible with an estimated 50–70% being bound to the cell surface or extracellular matrix following secretion [35]. *VEGF-A_165_* is the most abundantly expressed isoform and most potent initiator of angiogenesis [63]. In fact, VEGF-A_189_ has less angiogenic activity when compared to the VEGF-A_165_ and VEGF-A_121_ isoforms [64,65]. In addition, VEGF-A_165_ has increased mitogenic effects in comparison to VEGF-A_189_, which is caused by its heparin binding ability that allows the attachment to the neuropilin co-receptor [66]. 

It is hypothesized that many splice factors and signaling pathways are involved in the regulation of *VEGF-A* isoform expression. Indeed, CAPER-α, a transcriptional coactivator for steroid receptors has been shown to alter the ratio of *VEGF-A_189_* to *VEGF-A_165_* in Ewing cells [67]. Furthermore, an earlier study also showed that siRNA knockdowns of *CAPER-α* increased the *VEGF-A_121_*/*VEGF-A_189_* expression ratio in breast cancer cells [68]. Therefore, it can be concluded that CAPER-α is involved in the regulation of alternative splicing of these three isoforms of VEGF-A, and it is proposed that perhaps CAPER-α may also govern the selection of other VEGF-A isoforms. An additional layer of VEGF-A regulation has been uncovered, whereby microRNAs (particularly *miRNA-297* and *-299* in tumour-associated macrophages; and *miR-574-3p* in human myeloid cells) bind to the CA-rich element (CARE) in the 3′ UTR of *VEGF-A*, which results in the negative regulation of VEGF-A expression [69,70]. However, upon hypoxic insult, heterogenous nuclear ribonucleoprotein L (hnRNP L) is translocated from the nucleus to the cytoplasm, where it can interact with *VEGF-A* mRNA. More specifically, hnRNP L binds to CARE, thereby inhibiting the binding of the microRNAs to the sequence and subsequently induces the splicing of *VEGF-A* [69]. In addition, it has been shown that increased hnRNP L binding to the CARE element during hypoxia also reduces the binding of the GAIT complex to the GAIT element, which accordingly results in decreased translational repression of VEGF-A [71]. However, whether these mechanisms induce splicing for the expression of a certain isoform of *VEGF-A* is yet to be determined. 

Almost 20 years ago, another family of *VEGF-A* isoforms was discovered, which arise from alternative splicing of exon 8 (Figure 4A). The isoforms have the same number of amino acids, but alternative sequences in the C terminus [72]. So far, four alternative *VEGF-A_XXX_b* isoforms have been identified: *VEGF-A_165_b, VEGF-A_121_b, VEGF-A_189_b*, and *VEGF-A_145_b* [72,73,74,75]. It is generally accepted that the VEGF-A_xxx_b family of isoforms are inhibitors of angiogenesis, and the VEGF-A_xxx_a family promotes angiogenesis (Figure 3B), which arguably makes them the most intriguing splicing event of *VEGF-A*. In fact, it has been reported that VEGF-A_xxx_b expression is downregulated in cancer, diabetic retinopathy, Denys Drash syndrome and retinal vein occlusion, and upregulated in systemic sclerosis and asthma [38]. 

The aberrant splicing of *VEGF-A* exon 8 has been extensively investigated, and a few different signalling pathways have been reported to be involved in the production *VEGF-A_xxx_b*. In the first pathway, stimulation by growth factors (e.g., insulin growth factor (IGF)) promotes the phosphorylation of SRSF1 by SR protein kinase 1, which allows transportation of SRSF1 into the nucleus. SRSF1 then binds with the proximal splice site of exon 8 and results in the production of the *VEGF-A_xxx_a* isoform [76]. However, knockdown of SRSF1 and knockdown and inhibition of SRPK1 altered the splicing of VEGF-A mRNA to produce more of the *VEGF-A_xxx_b* isoform, and so it is thought that *VEGF-A_xxx_b* production is linked to the inhibition of this pathway [77]. 

The other three pathways described so far involve the selection of the distal splice site of VEGF-A exon 8, which results in production of the *VEGF-A_xxx_b* isoform. One pathway involves the stimulation of CLK1/4 phosphorylation of SRFS6 by TGF-β, which leads to distal splice site selection of *VEGF-A* exon 8 [78,79]. The remaining two pathways that have been uncovered to date involve *VEGF-A_xxx_b* splice site selection by SRSF2. One such pathway includes the wingless integrated 5a/orphan receptor tyrosine kinase-like receptor-2/Jun N-terminal kinase (Wnt5a/Ror2/JNK) dependent mechanism [80], and the other pathway involves E2F1 [81]. 

### 3.2. Vascular Endothelial Growth Factor Receptors (VEGFRs)

In order to facilitate a cellular response, a VEGF-A dimer binds to paired tyrosine kinase receptors, named vascular endothelial growth factor receptors (VEGFRs), which have been found in a range of cell types, such as endothelial cells [82], haematopoietic [83] and smooth muscle cells [84]. There are three types of VEGFRs: VEGFR1, VEGFR2 and VEGFR3, which are encoded by different genes; however, VEGFR1 and VEGFR2 share similar structures [85,86]. The VEGF-A family of isoforms bind to VEGFR1 and VEGFR2 to facilitate angiogenesis, whereas VEGFC/D isoforms bind to VEGFR3 to drive lymphangiogenesis for the formation of lymph vessels [87]. VEGFR1 has decreased kinase activity when compared to VEGFR2, despite having a higher affinity for the binding of VEGF-A, and so is highly regarded as a decoy receptor [88,89]. Therefore, most research into the roles of VEGFRs in angiogenesis have focused on VEGFR2.

An additional step is required to activate VEGFR2 signalling after VEGF-A has bound to the receptor. The binding of allosteric sites of VEGFR2 are necessary to cause a conformational twist of the receptor, which leads to its activation [34]. Similar structural arrangements are caused by VEGF-A_165_, VEGF-A_165_b and VEGF-A_121_, which ultimately allows the transduction of the signal [90]. VEGF-A_165_ has been shown to activate phosphorylation of VEGFR2, AKT and ERK the most, whereas VEGF-A_165_b activation of VEGFR2 results in diminished stimulation of angiogenesis [91,92,93], and it is for this reason the ‘anti-angiogenic’ label has been applied to VEGF-A_165_b. Most evidence points to a partial agonistic function for VEGF-A_121_ [93,94]; however, there have also been some cases where VEGF-A_121_ has shown full agonistic qualities. For example, several independent groups have shown that VEGF-A_121_ activation induced phosphorylation of VEGFR2, but at a reduced rate to VEGF-A_165_a [95,96,97]. In addition, when compared to VEGF-A_165,_ VEGF-A_121_ generated less motility and sprouting in human umbilical vein endothelial (HUVEC) cells [95]. Studies into the effect of VEGF-A_121_ on ex-vivo angiogenic sprouting and vascular permeability have demonstrated VEGF-A_121_ to have both equal and lower levels of generation to VEGF-A_165_ [34]. 

VEGFR2 signalling is complicated further by splicing of *VEGFR2*. *Endogenous soluble VEGFR2 (esVEGFR2)* [39] arises from partial retention of intron 13 and a consequential early termination stop codon. The resultant protein contains six out of the seven extracellular immunoglobulin-like domains and has a unique *c*-terminal sequence that does not exist in the canonical form of *VEGFR2*. It can be noted that there is another soluble variant of VEGFR2 which is formed through proteolytic shedding from the cell surface, which is a post-translational modification and is not formed from alternative splicing of *VEGFR2* [98]. Both of these variants act on the lymphangiogenic system through trapping of VEGF-C, which obstructs the activation of VEGFR3, resulting in the decrease of lymphatic endothelial cell proliferation [98]. Lymphangiogenesis in tumours promotes metastasis of malignant cancer cells. In accordance with this, researchers have observed a correlation between sVEGFR2 downregulation and progression of malignant neuroblastoma, which suggests a protective function for these variants in malignancies [39]. Another truncated *VEGFR2* isoform has been discovered in rat retina, which contains the entire extracellular domain and a part of the intracellular domain. Little is known about this isoform; however, it has confirmed VEGF-A activity and leads to increased cytoplasmic calcium and therefore is thought to be functional in the angiogenic system [99]. 

Although VEGFR1 is heavily regarded as a ‘decoy’ receptor for angiogenesis, it has been implemented in tumour progressive processes. In adults, VEGFR1 has been shown to activate inflammation, tumour growth, metastasis; increase chemotaxis of monocytes and has been linked to increased release of MMP9 from human vascular smooth muscle cells [99,100]. To date, four alternatively spliced variants of *VEGFR1* have also been discovered: *sVEGFR1-i13* (shown in Figure 4A), *sVEGFR1-i14, sVEGFR1-e15a* and *sVEGFR1-e15b* [101,102,103]. The *sVEGFR1-i13* and *sVEGFR1-i14* isoforms have extensions beyond exon 13 and exon 14, respectively, whereby *sVEGFR1-i13* arises from an extended read through of a potential splice site in exon 13 (Figure 4A), and *sVEGFR1-i14* arises from a skipped splice site. The *sVEGFR1-e15a* and *sVEGFR1-i15b* isoforms arise from the inclusion of an alternative exon in place of exon 15 (either exon 15a or 15b) [102,103]. Membrane-bound *VEGFR1* and the soluble isoforms share the first 13 exons, which code for the first six N-terminal immunoglobulin-like domains [103]. The soluble isoforms lack all the protein domains that are encoded after exon 13 in membrane-bound *VEGFR1*: the seventh immunoglobulin-like extracellular domain, the membrane-anchoring region, the juxtamembrane domain and the intracellular signaling tyrosine kinase domains [103]. This enables the soluble isoforms to have retained affinity for VEGF-A ligands, neuropilin-1 (NRP-1) co-receptor and extracellular heparin sulphate proteoglycans, but prevents signal transduction. In essence, sVEGFR1 isoforms trap VEGF-A and form inert heterodimers with VEGF receptors [39], which has caused them to be regarded as potent anti-angiogenic factors. Investigations into the mechanisms that control the splicing of VEGFR1 have been conducted, and numerous studies have provided evidence that hnRNP D plays a role in the regulation of *sVEGFR1-i13*. One such study executed in human dermal microvascular endothelial cells (HMVEC) unearthed two sequences within intron 13 that are responsible for the premature poly(A) signal found in *sVEGFR1*, and proposed that hnRNP D binds to an A-and-U-rich-element (AURE) sequence (AUUUA) located in intron 13 downstream of the *sVEGFR1* processing site [104]. Indeed, overexpression of hnRNP D was shown to significantly decrease the *sVEGFR1/mVEGFR1* (membrane-bound VEGFR1) ratio [104]. However, a later study conducted in human macrophage-like U937 cells provided evidence that hnRNP D increases the expression of *sVEGF1* mRNA with the use of peptides that were modelled on the regulatory domain of hnRNP D [105]. Furthermore, this finding was strengthened by the peptide-mediated suppressed expression of VEGF-A [105]. This discrepancy in the role of hnRNP D in *VEGFR1* splicing regulation could be due to a cell type-specific effect and/or additional modifications, such as the altered expression of other factors yet to be uncovered as being involved in the alternative splicing of *VEGFR1*. In addition, regulation of splicing events, particularly of genes that are of particular importance, can be very complicated with many molecules involved, and so it could be that there are other molecular mechanisms taking place that override the effect of hnRNP D. Indeed, there has been evidence of other molecules and pathways that regulate the expression of *sVEGFR1-i13*. Studies conducted in endothelial cells have revealed that the U2A65 splicing factor is implicated in the positive regulation of *sVEGFR1-i13* [106], whereas JuMonJi Domain containing protein-6 (JMJD6) and NOTCH1 have been shown to be negative modulators of the isoform [106,107]. Moreover, a recent study conducted in squamous lung carcinoma cells has shown evidence for a more accomplished mechanism, whereby VEGF-A_165_ promotes the expression of SOX2, which directly binds to the promoter of SRSF2 and induces transcription [101]. The SRSF2 splice factor promotes splicing of *VEGFR1* towards the *sVEGFR1-i13* isoform [101]. 

### 3.3. Neuropilins (NRPs)

NRPs are expressed in many distinct cell types, including neurons, blood vessels and immune cells [108]. NRPs are transmembrane co-receptors that function in both VEGF signalling (through dimerization with VEGFR2) and semaphorin/plexin signalling, which implicate NRPs in both angiogenesis and lymphangiogenesis, and neuronal tissue pattern development, respectively [109]. For the basis of this review, the role of NRPs in axonal development will be kept brief, as this subject matter is beyond the scope of this review. However, it can be noted that the semaphorin-3 (SEMA3) family of glycoproteins binds to NRP receptors to facilitate axonal guidance.

The structure of NRPs consists of various domains with different functions. The a1 and a2 domains are involved in binding to SEMA3, and the b1 and b2 domains are required for the binding of VEGF-A; in particular, VEGF-A_165_, VEGF-A_121_ and VEGF-A_189_ have been shown to bind to NRP1 [108]. In fact, VEGF-A_165_ has been shown to be paramount for the complex formation between VEGFR2 and NRP1 [110]. VEGF-A_165_ can also bind to NRP2, however, with a 50-fold less than NRP1 [111]. Interestingly, exon 8 of VEGF-A has been shown to be crucial for binding to NRPs, and consequently VEGF_165_b has been shown to be unable to bind to NRPs [112]. In addition to VEGF-A, the b1 and b2 domains of NRP1 have also been shown to bind to other angiogenic-associated factors, including: VEGF-B, VEGF-C, VEGF-D, placental growth factor-2 (PLGF2), hepatocyte growth factor (HGF), fibroblast growth factors (FGFs), and transforming growth factor beta 1 (TGFβ1) [108], which further corroborates that NRP1 plays a key role in angiogenesis. 

*NRP1* mRNA consists of 17 exons and to date, six alternatively spliced isoforms of *NRP1* have been discovered. The first to be discovered were *s_11_NRP1* and *s_12_NRP1*, which are composed of the first 11 and 12 exons of *NRP1*, respectively [40,113]. An additional two more isoforms were later discovered: *s_III_NRP1* and *s_IV_NRP1. s_III_NRP1* is formed from exons 1-9 and exon 12, with cassette exon skipping of exons 10 and 11 [114]. The structure of *s_iv_NRP1* is similar to that of *s_III_NRP1*, however, it also retains exon 10 [114]. All four of these isoforms are soluble and do not contain the C-terminal transmembrane and cytoplasmic domains. The resulting proteins formed are able to sequester NRP1 binding factors, but are unable to transmit signals, and therefore act as antagonists of NRP1 signalling [40]. Hence, these soluble isoforms of *NRP1* have been associated with anti-angiogenic and anti-tumourigenic functions. Another splice variant of *NRP1*, named *NRP1-ΔE16*, is formed by the skipping of exon 16 and the addition of an arginine (AAG) [41]. Not much is documented on this isoform, as it was not found to have any functional differences to full length *NRP1*. 

More recently, a further splice isoform of *NRP1*, *NRP-1Δ7*, was discovered [42]. This isoform is formed by the deletion of just seven amino acids in exon 11 caused by the utilization of an alternative donor site located 21 bases upstream of the conventional splice site (Figure 4A). The seven amino acids that are deleted in *NRP1-Δ7* omit a coding sequence that contains two aspartic residues that are located two residues downstream of the glycosylation site and are required for efficient glycosylation [115]. Glycosylation is a post translational modification that can alter the structure and function of a protein. NRP1 is reported to be glycosylated at Serine-612, whereby long chains of oligosaccharides are formed. Various different chains can assemble on NRP1, which are dependent on the cell type. For example, in smooth muscle cells, chondroitin sulfate is added, whilst in endothelial cells heparin sulfate is assembled on the glycosylation site of NRP1 [116]. The number and nature of the added oligosaccharide chain(s) can modulate the function of a protein. Indeed, it has been reported that mutated Serine-612 prevents glycosylation at the site, increased cell invasion in human glioblastoma cells [117] and reduced fibronectin fibrillation in a human hepatic cell line [118]. Further studies conducted in modified prostate (PC3) and breast (MDA-MB-231) cancer cells that expressed either recombinant full length NRP1 or NRP1-Δ7, showed that NRP1-Δ7 has anti-tumorigenic properties, including decreased proliferation, migration and anchorage-dependent growth whilst the overexpression of full length NRP1 had the opposite effect [42]. Therefore, NRP1-Δ7 is considered to function in the inhibition of angiogenesis and tumorigenesis and could have therapeutic potential. 

NRP2 functions in lymphangiogenesis through dimerization with VEGFR3 and activation by VEGF-C [119]. Indeed, *NRP2* knockout mice presented with impaired lymphangiogenesis [120]. In addition, NRP2 has been implicated in tumour metastasis. This was shown using a monoclonal antibody to inhibit VEGF-C binding to NRP2, which in turn decreased metastasis [121]. Recently, a new splice isoform of *NRP2* named *s_9_NRP2* has been discovered and is thought to antagonize VEGF-C/NRP2 signalling [122]. The isoform is formed from the retention of an intron between exons 9 and 10 that encodes a premature stop codon in the b2 domain. This results in an unpaired cysteine residue that forms a disulphide bridge with an unpaired cysteine from another s_9_NRP2-producing dimer. There has been evidence to show that s_9_NRP2 inhibits the binding of VEGF-C to NRP2, however, with no effect on the interaction of VEGF-C to VEGFR-3 [122]. Further research is required to establish whether the isoform also inhibits the formation of metastasis. 

### 3.4. Fibroblast Growth Factor Receptors (FGFRs)

There are at least five types of FGFRs named FGFR 1–5 [123]; FGFR 1-3 have a similar structure, whereas FGFR4 lacks the transmembrane and intracytoplasmic domains. FGFR5 and an additional receptor have also been proposed, designated as FGFR6 [124]. FGFRs have the same protein structure that is generally found in receptor tyrosine kinases and are involved in the regulation of key biological cellular processes, such as proliferation, differentiation, migration and survival [125]. There are a number of different types of alternatively spliced isoforms that can arise in *FGFR*s, which will be discussed below. 

The most described alternative splicing event in the literature is generated from the use of alternative exons in the c-terminus of IgIII of *FGFRs1-3* [43]. The IgIII domain is comprised of three domains: IIIa, IIIb and IIIc, which are encoded by exons 7–9, respectively. The alternative use of exons 8 and 9 form the IIIb and IIIc isoforms for *FGFR1-3* (Figure 4A), which have different ligand affinities and tissue specificities. i.e., the IIIb isoform is predominantly found in the epithelial tissues, whereas the IIIc isoform is expressed in the mesenchymal tissues. The regulation of this splicing switch is therefore strongly implicated in epithelial-mesenchymal-transition (EMT). A number of RBPs have been implicated in the regulation of splicing of the c-terminus of *IgIII*. ESRP1 and ESRP2 promote the expression of the *IIIb* isoform, and hnRNP F/H, K and forkhead box-2 (FOX-2) silence the *IIIc* isoform. On the other hand, hnRNP A1 and PTB are associated with silencing of *IIIb* [126]. Furthermore, hnRNP M has been shown to bind to ISE/ISS-3 of *FGFR2* and promote exon IIIc skipping [127]. Another more detailed splicing mechanism for the regulation of *IIIb* and *IIIc* splicing involves RBM4 and nPTB [128]. In short, RBM4 promotes the expression of the *IIIb* isoform, whereas nPTB induces exon IIIc inclusion. Interestingly, the RBM4 splice factor is also implicated in promoting exon 10 skipping of *PTB* to produce the *nPTB* splice variant [128], and this therefore shows the extent that splicing mechanisms are tightly controlled. The splicing regulation of *FGFR2* is a prime example of how RNA-binding proteins work in combination to bring about a splicing profile. A mesenchymal phenotype is linked to the promotion of tumorigenesis, and aberrant expression of the IIIb and IIIc isoforms has been documented in animal models of bladder cancer and epithelial cell tumours [129,130,131]. Furthermore, elevated expression of both the IIIc and IIIb isoforms has been shown in a variety of cancerous cells and tissues [43]. However, expression of FGFR-2 IIIb in oesophageal and colorectal cancers was associated with better differentiation and decreased expression of the isoform correlated with proliferation, invasion and poor prognosis [44]. Furthermore, the loss of the IIIb isoform and increase in IIIc expression has been associated with a more aggressive cancer phenotype [44]. In light of the above, it could be concluded that the IIIc isoform is a tumour promoter, and IIIb acts as a tumour suppressor; however, increased expression of FGFR-2 IIIb has also been linked to cell transformation and tumour progression in some cancers, and so the precise role of FGFR-2 IIIb in cancers remains controversial [44]. For example, in gastric cells, FGFR-2 IIIb expression was associated with cellular invasion of the gastric wall; however, decreased expression of FGFR-2 IIIb in gastric cells was also linked to proliferation, invasion and poor prognosis [44]. Furthermore, FGF-7 and FGFR-2 IIIb have been implicated in tumour angiogenesis through inducing an increase in VEGF-A expression in colorectal and pancreatic cancers [132,133]. This is a prime example of how there is a lot of cross-talk between different arms of angiogenesis signalling, and therefore associates FGF-7 and the FGFR-2 IIIb isoform in particular in the promotion of angiogenesis. 

The next splicing event of *FGFRs* that will be discussed is the inclusion or exclusion of exons that encode the IgI and acid box domains, which are formed from exons three and four respectively in *FGFRs 1–3*, and exons two and three respectively in *FGFR4* [134,135]. Both domains are involved in the auto-inhibition of FGFRs [43]. In fact, isoforms that do not contain these domains have heparin sulfate proteoglycan attachments, which strengthens the affinity for binding to ligands and enhances signalling transduction of FGFRs [136]. The excision of the IgI domain produces an isoform that has two loops in the extracellular domain (formed by IgII and IgIII), named *FGFRβ*, whereas the inclusion of the exon produces *FGFRα*, which has three loops in the extracellular domain [137]. Therefore, the FGFRβ isoforms, which do not contain the auto-inhibitory IgI domain, have a higher affinity for FGFs and enhanced signalling [45,46]. Indeed, FGFR-1β has been shown to increase proliferation through having a stronger affinity for FGF1 and displaying augmented signalling [43]. Furthermore, an elevation in the expression ratio of *FGFR1β:FGFR1α* has been linked to tumour progression [43]. Recently, the polypyrimidine tract-binding protein 1 (PTPBP1) splicing repressor has been implicated in the regulation of alternative splicing of *FGFR1α* and *FGFR1β*. The study conducted in breast cancer cells revealed that PTBP1 represses the splicing of *FGFR1β* [138]. However, PTBP1 has also been reported to induce α exon deletion through the binding of intronic splicing silencer sequences that flank the α exon resulting in the positive regulation of *FGFR1β* [139]. Therefore, further investigation is required to clarify the precise role of PTBP1 in the regulation of *FGFR1α/FGFR1β* splicing. In addition, the SRSF6 splicing factor has been implicated in the exclusion of the α exon, which has been attributed to the presence of two exonic splicing enhancers located in the α exon, which can be bound by SRSF6 to facilitate the exclusion of the α exon [140]. Indeed, knock downs of SRSF6 resulted in the increased expression of *FGFR1α* [140]. 

Another class of isoforms are formed from the skipping of exons 8, 9 and 10. This region encodes the transmembrane domain, which is responsible for anchoring FGFRs to the cellular membrane. Therefore, this alternative splicing event produces soluble FGFR receptors, which can be found in other locations in the cell other than the cell membrane. For example, a soluble isoform of FGFR3, which arises from the deletion of exon 8 as well as exon 7, has been shown to be predominantly found in the nucleus of breast epithelial cells [45]. Although the precise functions of these isoforms are yet to be determined and may be diverse depending on the splicing events that exist in other regions of the sequence, it can be proposed that many of the soluble isoforms act as an antagonist through the binding of ligands without the activation of signal transduction, which is true of other soluble tyrosine kinase receptors. Indeed, a soluble isoform of FGFR4 has been shown to diminish FGF1 signalling in human Michigan Cancer Foundation-7 (MCF-7) breast cancer cells [43]. 

A further group of FGFR splice isoforms arise from the inclusion of differing carboxy-terminal sequences, termed *C1*, *C2* and *C3*. The distinct sequences produce proteins that have differential retention of tyrosine residues, which function as docking sites for cytoplasmic signalling proteins and are also involved in autophosphorylation of the receptor [47]. The C3 variant has been documented to be the most transforming of the isoforms when compared to C2, which has moderate transforming activity, and C1 which has weak activity. The increased transforming activity elicited by the C3 isoform strongly suggests a role for these variants in oncogenesis. In support of this, elevated expression of the C3 isoform has been detected in gastric cell lines and in human breast cancer cell lines [47].

Another type of splice isoform that arises in *FGFRs* is produced via the inclusion or deletion of six nucleotides (GTAACA), known as the VT motif which codes for two amino acids (valine and threonine) in the juxtamembrane region of FGFR1-3 [141]. The inclusion or exclusion of the VT motif affects the signalling capability of the receptor, as the exclusion of the VT motif prevents the binding of effector molecules. In fact, isoforms that contain the VT motif are able to activate the Ras/MAPK signalling pathway [48]. Therefore, as one of the major downstream pathways of FGFR is affected, it has been postulated that the absence of the VT motif could have major implications for the function of FGFR. However, other signalling pathways that rely on receptor kinase activity for activation remain unaffected by the exclusion of the VT motif, such as the downstream effectors of phospholipase C-γ (PLCγ) [141]. Therefore, the inclusion or exclusion of the VT motif may selectively activate different signalling mechanisms in varied cell types. 

### 3.5. Vasohibins

Vasohibin-1 is a negative regulator of angiogenesis that is typically expressed in endothelial cells and is regulated by VEGF-A and FGF-2 [51]. Whilst vasohibin-1 inhibits VEGF-stimulated angiogenesis, it does not inhibit the phosphorylation and subsequent activation of VEGFRs; however, vasohibin-1 does inhibit FGF-2-activated angiogenesis [142]. In fact, it was also shown that vasohibin-1 is activated by VEGFR-2 and its downstream effector, protein kinase C-δ (PKC-δ), in endothelial cells resulting in the inhibition of angiogenesis [51]. PKC-δ is also involved in the induction of vasohibin-1 by FGF-2, which further implies that this factor is imperative to the regulation of vasohibin-1 expression [51]. Vasohibin-1 has also been shown to regulate the hypoxia inducible factor-1α (HIF-1α) pathway through inducing the degradation of HIF-1α by prolyl hydroxylase, which suggests that hypoxia and vasohibin-1 act in concert to regulate one another [143]. Therefore, the expression of vasohibin-1 is highly regulated in a number of ways. 

Later, another gene which shared 52.5% homology with vasohibin-1 was discovered and designated vasohibin-2 [144]. Human vasohibin-2 was shown to also exert anti-angiogenic activity; however, unlike vasohibin-1, vasohibin-2 was not found to induce VEGF–A or FGF-2. Furthermore, vasohibin-2 is not regulated by cytokines or growth factors [144]. However, a microRNA (*mir-200b*) has been shown to target and therefore diminish the expression of vasohibin-2 [144,145]. 

Studies of multiple collections of microarray data has revealed that the two genes have entirely different profiles of co-expression with other genes, suggesting that vasohibin-1 and vasohibin-2 have distinct functions [144]. In accordance, vasohibin-1 is expressed in endothelial cells of newly formed blood vessels behind the sprouting front and acts to halt angiogenesis, whereas vasohibin-2 is expressed in mononuclear cells that help form the sprout and promotes angiogenesis [144]. Indeed, vasohibin-2 has been linked to tumour growth through the stimulation of angiogenesis [145,146], whereas vasohibin-1 has been shown to be an effective treatment for angiogenesis in various animal models, including diabetic nephropathy, pulmonary fibrosis, ocular angiogenesis and cancers [144]. 

The two main isoforms of *vasohibin-1* that have been documented in the literature are the 365aa isoform, known as *vasohibin-1A* (*VASH1A*) formed by seven exons, and the 204aa isoform, known as *vasohibin-1B* (*VASH1B*), which consists of four exons [50] (Figure 4A). The proteins generated from these isoforms have the same N-terminus, which contains a nuclear localisation signal, but a different C-terminus, which is thought to be important for anti-angiogenic activity [49]. Interestingly, a study of in vivo blood vessel growth conducted in chicken chorioallantoic membrane showed that whilst VASH1A did not inhibit vessel growth, the vessels were shown to be irregular; however, VASH1B inhibited vessel growth strongly [49]. Furthermore, in vitro studies of angiogenesis showed that VASH1B inhibited endothelial cell growth, migration and tube formation [49]. VASH1A on the other hand, only had a significant effect on migration in vitro; in fact VASH1A significantly increased endothelial cell migration [49]. Moreover, the VASH1B isoform was shown to induce apoptosis in endothelial and fibroblast cells, whereas VASH1A was not shown to only induce apoptosis in fibroblasts [49]. A later study was able to explain the differences in functions exerted by the two vasohibin isoforms [50]. It was shown that VASH1A promoted normalisation of abnormal tumour blood vessels, whereas VASH1B induced autophagy, which promotes either cellular survival or cellular death in order to maintain homeostasis. Therefore, the same group investigated the effects of combination treatments that included both isoforms and found that the combination treatments had the highest anti-tumour effects with marked vascular normalisation when compared to treatment with either isoform alone [50]. Therefore, combination treatments with VASH1A and VASH1B may be useful in antiangiogenic treatments, as the existing vasculature can be normalised by VASH1A and pruned by VASH1B. 

The alternative splicing of *vasohibin-2* is less documented. *Vasohibin-2* contains 11 exons which are alternatively spliced to produce transcripts that are*: 355aa, 311aa, 290aa, 156aa, 117aa* and *104aa*. The largest isoform (355aa) has been shown to be predominantly expressed in HUVEC cells, and the 290aa splice variant has been shown to have antiangiogenic activity [51]. The function of the other isoforms have not yet been classified. As mentioned in the beginning of this section, vasohibin-2 was found to be involved in the stimulation of angiogenesis and is highly expressed at the sprouting front. The drive of angiogenesis by vasohibin-2 was linked to high expression of the protein in mononuclear cells at the sprouting front, whereas anti-angiogenic activity of vasohibin-2 was identified in endothelial cells. This suggests that perhaps vasohibin-2 has distinct functions in different cell types. Another reason for this discrepancy could be that splice isoforms of vasohibin-2 whose function has not yet been determined could have opposing functions. 

### 3.6. Hypoxia Inducible Factor-α (HIF-α)

Hypoxia is a feature of many diseases, such as cancers whereby inadequate vasculature causes pockets of low oxygen. This exerts a pro-apoptotic response in some cancer cells [147], as they do not receive the oxygen they require for survival. Therefore, hypoxia elicits various responses including the promotion of angiogenesis in order to provide cells with the oxygen that they need. The most established signalling pathway that is induced by hypoxia is the hypoxia inducible factor-α (HIF-α) pathway. In short, HIF-α forms a dimer with HIF-β, also known as aryl hydrocarbon receptor nuclear translocator (ARNT), which then binds to HIF-response cis-elements to facilitate the transcription of target genes, including *VEGF-A*. However, in normoxic conditions, prolyl hydroxylases (PHD) interacts with HIF-α, allowing the von Hippel-Lindau (VHL) protein to bind and target the degradation of HIF-α by the proteasome thereby preventing any transcription of HIF-targeted genes [8]. There are three homologous of *HIF-α* and *HIF-β* that are designated *HIF-1α* and *HIF-1β*, *HIF-2α* and *HIF-2β*, and *HIF-3α* and *HIF-3β* [55]. The HIF-1α and HIF-2α subunits share similar domain assemblies and undergo proteolytic regulation [55]. However, HIF-3α has been generally considered as a negative regulator of HIF-1α, but not HIF-2α [148]. The most studied of these are the HIF-1α subunits.

Several splice isoforms of *HIF-1α* have been described in the literature, which arise from cassette exon skipping: *HIF-1α Δ11*, *HIF-1α Δ12*, *HIF-1α Δ11&12*, *HIF-1α Δ14*, and *HIF-1α^417^* [53,54,149]. The splicing scheme for *HIF-1α Δ14* is shown in Figure 4A. Most of these isoforms lack exons that encode parts of the oxygen-dependent degradation domain which is targeted by prolyl hydroxylases (PHDs), or C-terminus transactivation domains, which are required for transcription of HIF-targeted genes [149]. As most of these isoforms have reduced or completely abolished HIF function and have a dampening effect on the initiation of HIF-activated transcription of target genes, it is therefore predicted that these isoforms are stable in normoxia in order to maintain negative regulation of HIF signalling in the presence of oxygen. In accordance with this, a study found that the HIF-1α Δ14 isoform is expressed at higher levels than the canonical isoform during normoxia [54]. During hypoxia, however, both the canonical isoform of HIF-1α and HIF-1α Δ14 were able to dimerise with HIF-1β and activate the *VEGF-A* promoter [54]. However, HIF-1α Δ14 was shown to have a three-fold less potency than the canonical form of HIF-1α [54]. Furthermore, HIF-1α Δ12 and HIF-1α Δ11&12 both block the dimerization of HIF-1α and HIF-1β by isolating HIF-1β to the cytoplasm [53]. Therefore, these two isoforms act as dominant-negative regulators of HIF-1 transcription. The skipping of exon 10 produces the *HIF-1α^417^* splice variant, which causes a frame-shift and results in a truncated protein of 417 amino acids in length [53]. The resultant protein lacks a transactivation domain and so was thought not to act as a principal transcription factor. However, further experiments showed that the splice variant amplified HIF-1β-mediated transcription of the *erythropoietin* (*EPO*) reporter gene [53], which shows another mechanism in which HIF target genes are transcribed. In contrast to the other isoforms, there is evidence to show that HIF-1α Δ11 promotes tumorigenesis through the enhancement of HIF-1 activity. Indeed, overexpression of HIF-1α Δ11 increased tumour growth in vivo [52]. An additional alternative splicing event involves the insertion of three bases (TAG) between exons 1 and 2 of HIF-1α, which substitutes an asparagine in place of a lysine and inserts an additional arginine [44]. This significance of this isoform remains to be elucidated. Another splice variant of HIF-1α arises from an alternative upstream translational start site, which produces a protein that has an N-terminus that is 24 amino acids longer than canonical HIF-1α named *HIF-1α Alt1* [150]. 

*HIF-3α*, the dominant negative regulator of HIF-1α, has also been reported to undergo alternative splicing [47,151]. The most established alternative splice isoform of *HIF-3α* is *inhibitory Per/Arnt/Sim* (*IPAS*), which is formed through multiple alternative splicing events and is inducible by hypoxia (Figure 4B). *IPAS* shares exons 2, 4 and 5 with the canonical *HIF-3α*; however, *IPAS* also contains an alternative exon upstream of exon 1 (designated 1a), as well as an additional exon before exon 4 (designated exon 4a) and exon 16. The use of alternative splice sites in exons 3 and 6 are also present in *IPAS* [151]. The inclusion of exon 4a, together with the use of the alternative 3′ splice site of exon 3, results in a frame shift. The resultant protein lacks transactivation domains and so cannot initiate transcription [53]. IPAS dimerises with HIF-α subunits, thereby preventing the transcription of HIF target genes, such as *VEGF-A* [55]. Indeed, in cornea epithelium, IPAS was found to negatively regulate *VEGF-A* gene expression and therefore, was shown to indirectly dampen angiogenesis [56]. Another splice variant of *HIF-3α* that is generated by the inclusion of intron 7 and known as *HIF-3α4* is reported to be similar to *IPAS* [55,152]. HIF-3α4 forms a complex with HIF-1α, which prevents the binding of HIF to hypoxia response elements, and subsequent transcription of target HIF genes. Indeed, inducible HIF-3α4 was shown to hamper angiogenesis and proliferation [55]. 

### 3.7. Angiopoietin

Angiopoietins and Tie receptors are involved in endothelial cell survival and vascular maturation. There are four members of the angiopoietin family, designated angiopoietin 1–4 (Ang-1-4) [153]. Ang-1 and Ang-2 are the most characterised in the literature. They share around 60% homology and both bind to the same site of the TEK receptor tyrosine kinase-2 (Tie-2) receptor with similar affinity [153]. Ang-1 is expressed by smooth muscle cells and perivascular cells, whereas, Ang-2 is primarily expressed by endothelial cells and is stored in Weibel-Palade bodies within the cells [153,154]. Upon cytokine activation of the endothelium, Ang-2 is released from the Weibel-Palade bodies through a neuroligin-2-dependent mechanism and binds to the Tie-2 receptor [155]. Originally, it was thought that Ang-1 acted as an agonist of the Tie-2 receptor and Ang-2 was an antagonist of the Tie-2 receptor, as it has been shown to bind to Tie-2 but did not induce autophosphorylation of the receptor [156]. However, further investigations have shown that Ang-2 can also act as an agonist of the Tie-2 receptor. A range of solutions have been proposed to explain the contrasting functions assigned to Ang-2, such as cell type-specific effects, Ang-2 stimulation duration, the presence of co-receptors (e.g., Tie-1), and the presence or absence of other molecules [153]. Indeed, it has been shown in chicken testis that Ang-2 induces angiogenesis in the presence of VEGF-A, whereas in the absence of VEGF-A, Ang-2 was associated with vascular regression [58]. Alternative spliced variants of Ang-2 with varied functions could also account for the discrepancies for functional assignment of Ang-2. 

The first discovered isoform was the *Ang-2B* variant, which was detected in chicken testis, and arises from the inclusion of exon 1B instead of exon 1 [58]. The study found that the canonical isoform (assigned Ang-2A) had a higher expression than the alternatively spliced Ang-2B isoform in immature testis and fully regressed testis, whereas the Ang-2B isoform was expressed more highly in adult quiescent testis than the Ang-2A isoform. Therefore, the study theorised that although a precise function could not be assigned to the Ang-2B isoform, there was an indication that Ang-2A may be involved in vascular remodelling in immature testis and fully regressed testis, whereas the Ang-2B isoform may be involved in inactivating the vasculature in quiescent testis [58]. Soon after the discovery of the *Ang-2B* isoform, another splice variant was discovered in humans, which arises from the skipping of exon 2, named *Ang-2_443_* (and also known as *Ang-2C*) [59,157,158] (Figure 4A). In likeness to Ang-2, the splice variant was also shown to bind to Tie-2 but did not stimulate Tie-2 phosphorylation, thereby acting as an antagonist of Tie-2 signalling activation [59]. Therefore, there is an indication that it may be involved in the regulation of angiogenesis. Furthermore, moderate Ang-2_443_ expression was also found in the C33A and CaSki cervical carcinoma cell lines and the isoform was also detected in primary tumour tissues, hemangioma and breast carcinoma, suggesting that the isoform may be involved in tumorigenesis of non-endothelial tumour cells [59]. Moreover, high expression of the isoform was detected during early macrophage differentiation, suggesting a role in the regulation of inflammatory processes [59]. *Ang-1* has four reported splice variants, which are named according to their size; however, the type of splicing that generates each of them is not clear. Albeit, there has been some research into the function of the isoforms. Both 1.5 kb (full-length Ang-1) and 1.3 kb bind to Tie-2 and induce its autophosphorylation, whereas the 0.9 kb and 0.7 kb splice variants also bind to the Tie receptor but are unable to induce its autophosphorylation [57]. Therefore, the 0.9kb and 0.7kb splice variants act as inhibitors of Ang-1 signalling. 

Several studies have implicated Ang-1 and Ang-2 in the promotion and inhibition of tumourigenesis [159,160,161,162]. There could be a few reasons for the discrepancies in tumorigenic function of Ang-1 and Ang-2, including: the effects of Ang-1 or Ang-2 may be cell-type specific; both angiopoietins may be increased and so determining the function of one may be construed as the function exerted by the other highly expressed angiopoietin [153]; or the variation could be due to differential expression of Ang-1 or Ang-2 splice variants. 

## 4. Manipulation of Alternative Splicing as a Potential Therapy for Angiogenic Associated Diseases

Many genes are alternatively spliced to produce splice isoforms that act as negative regulators. Therefore, there is an interest in exploiting this system in order to switch splicing profiles in disease to favour certain isoforms for therapeutic benefit. In other words, the general idea is to downregulate the expression of disease-causing isoforms to favour the expression of isoforms that do not promote disease progression.

One of the most common methods for achieving this is through the use of splice switching oligonucleotides (SSOs), which function through binding to *cis*-elements, and inhibiting the binding of splice factors at a specific splice site. In 2016, the FDA approved the first exon skipping SSO to be used in humans known as eteplirsen for the treatment of Duchenne Muscular Dystrophy (DMD) [163,164].

Another method of altering splicing profiles is through the manipulation of molecular mechanisms that regulate the splicing of a given gene by using small molecule compounds. An example of this is SPHINX, which is a small molecular compound that inhibits the SRPK1 splice factor kinase. SRPK1 phosphorylates and activates the SRSF1 splice factor, which is involved in the induction of the *VEGF-A_xxx_* pro-angiogenic isoform. However, when SRPK1 is inhibited by SPHINX, SRSF1 is not activated and the splicing profile is switched to favour the anti-angiogenic *VEGF-A_xxx_b* splice isoform. SPHINX has been shown to successfully reduce angiogenesis in several mouse models, including: ocular neovascularization [165], melanoma xenografts [166] and orthotopic prostate cancer [167]. Recently, a more potent form of SPHINX has been developed, named SPHINX31, which has a reported IC_50_ of 6nM [168]. In a similar thread, recently, another group uncovered the SRPK1-inhibitory function of an existent FDA-approved drug originally described as an ALK1 inhibitor called Alectinib, which had a half maximal inhibitory concentration (IC_50_) for SRPK1 kinase activity of 11nM [169]. Therefore, the next few years should yield some exciting findings and possible options for splicing-targeted angiogenic therapy.

## 5. Conclusions

Angiogenesis is a complicated process that is regulated by a variety of factors, and alternative splicing of angiogenic genes adds another layer of regulation to the angiogenic process. There are many splice isoforms that have a negative effect compared to the canonical isoform, which allows for careful regulation of the complicated angiogenic process. The importance of careful regulation is shown in pathologies whereby angiogenesis is dys-regulated, such as cancer, diabetic nephropathy, rheumatoid arthritis and endometriosis. Indeed, there have been many studies that have identified aberrant expression of angiogenic splice isoforms in cancers as described in this review. It is hoped that future studies will uncover more of the mechanisms associated with the generation of the splice isoforms mentioned for therapeutic benefit. Furthermore, the identification of additional splice isoforms of angiogenic-associated genes will allow more scope for the development of therapeutic drugs. 

## Figures and Tables

**Figure 1 ijms-20-02067-f001:**
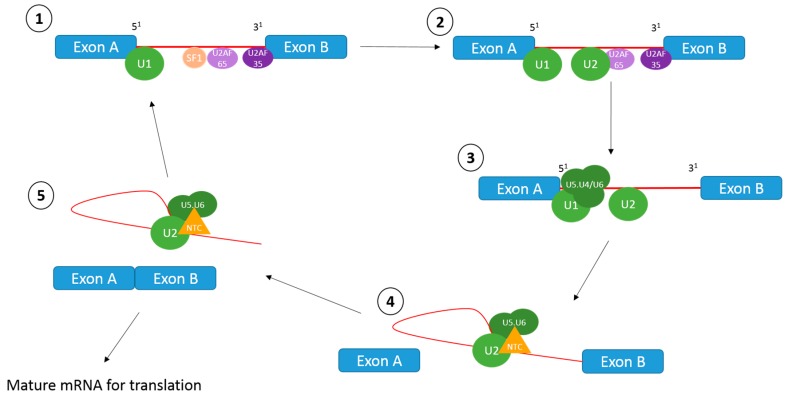
The splicing reaction. U1 snRNP binds to the CAG GURAGU consensus sequence, SF1 binds to the branch point sequence and U2AF35 binds to the 3′ splice site. The downstream polypyrimidine sequence of the branch point is bound by the U2AF65 subunit. SF1 is displaced by U2 and the U5. U4/U6 tri-snRNP is recruited to U1. Conformational and compositional rearrangements occur, which results in the release of U1 and U4, the addition of the NineTeen Complex (NTC), and the first transesterification reaction. A second transesterification reaction occurs which releases the lariat. Exons are ligated together and continue to translation. Introns are degraded and snRNPs are reprocessed for other splicing reactions.

**Figure 2 ijms-20-02067-f002:**
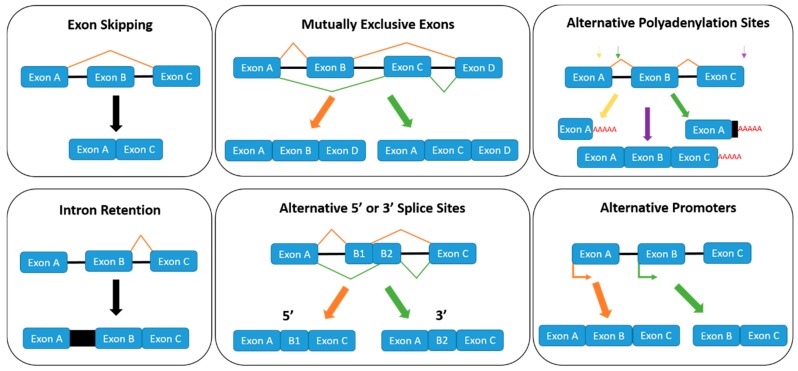
Different alternative splicing mechanisms; exon skipping, mutually exclusive exons, alternative polyadenylation sites, intron retention, alternative 5′ or 3′ splice sites and alternative promoters. The coloured splicing patterns correspond to the coloured arrows.

**Figure 3 ijms-20-02067-f003:**
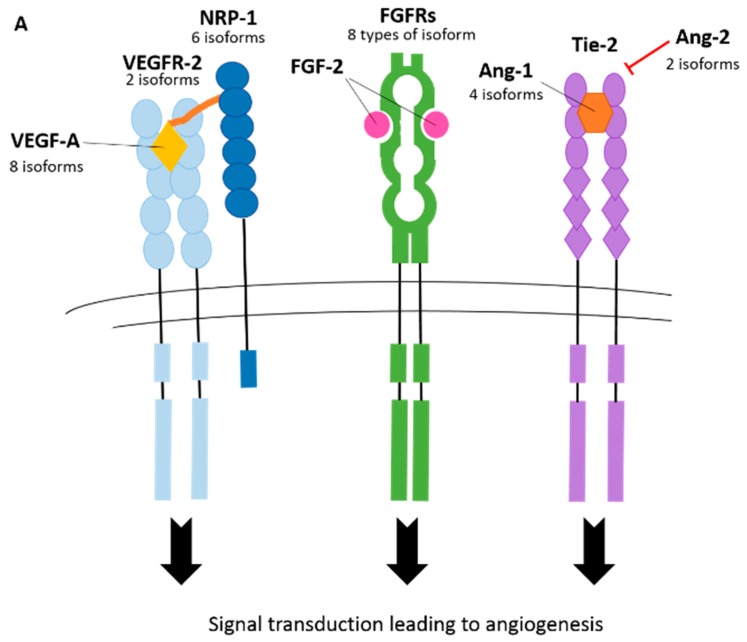
Alternative splicing of tyrosine kinase receptors and ligands that modulate angiogenesis. (**A**) Tyrosine kinase receptors: VEGFR-2 (with NRP-1 co-receptor), FGFRs and Tie-2 are shown with corresponding major ligands: VEGF-A, FGF-2, Ang-1, respectively. The number of currently known isoforms is indicated. (**B**) The effect of canonical VEGF-A_xxx_ and alternatively spliced VEGF-A_xxx_b on the activation of the VEGFR-2 receptor and downstream signalling cascade leading to angiogenesis. Canonical VEGF is proangiogenic, whereas VEGF-A_xxx_b acts a partial agonist of the receptor and does not induce angiogenesis.

**Figure 4 ijms-20-02067-f004:**
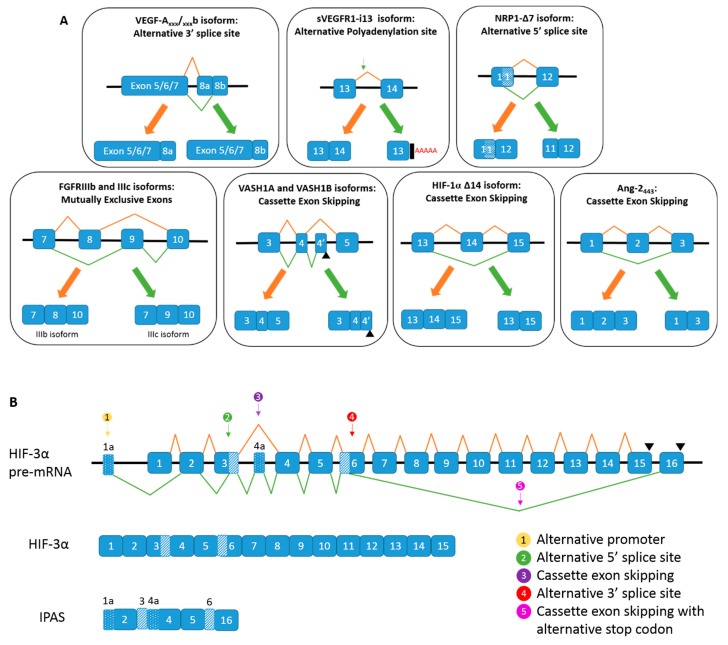
(**A**) Splicing schemes of major isoforms associated with angiogenesis. The coloured splicing patterns correspond to the coloured arrows. Premature stop codon shown by 
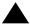
. (**B**) Alternative spicing of *HIF-3**α*, and the generation of *IPAS* mRNA.

**Table 1 ijms-20-02067-t001:** Splice isoforms of key angiogenic genes and their known functions.

Gene	Splice Variants	Function
*VEGF-A*	VEGF-A_111_	Proangiogenic. Diffusible [33,34]
	VEGF-A_121_	Proangiogenic. Diffusible. Implicated in tumorigenesis [33,35,36]
	VEGF-A_145_	Proangiogenic. Bind to cell surface and extracellular matrix [33,34]
	VEGF-A_165_	Proangiogenic. Moderately diffusible. Implicated in tumorigenesis [33,36]
	VEGF-A_189_	Proangiogenic. Implicated in tumorigenesis [33,36,37]
	VEGF-A_206_	Proangiogenic. Strongly bind to cell surface and ECM [33,37]
	VEGF-A_183_	Proangiogenic [33]
	VEGF-A_xxx_b	Anti-angiogenic. Downregulated in cancer, diabetic retinopathy, Denys Drash syndrome, retinal vein occlusion. Upregulated in systemic sclerosis and asthma [33,38]
*VEGFR1*	sVEGFR1	Potent anti-angiogenic [39]
*VEGFR2*	sVEGFR2	Decreases lymphangiogenesis. Downregulated in neuroblastoma [39]
	esVEGFR2	Decreases lymphangiogenesis [39]
*NRP-1*	s_11_NRP1, s_12_NRP1, s_III_NRP1, s_IV_NRP1	Soluble isoform. Antagonists of NRP1 signalling. Anti-angiogenic and anti-tumorigenic [40]
	NRP1-ΔE16	No functional difference to full length NRP1 [41]
	NRP1Δ7	Affects glycosylation status of NRP1. Anti-tumorigenic in prostate cancer and breast cancer cells [42]
*FGFRs*	IIIb	EMT. Found in epithelial tissues. Evidence as a tumour suppressor and as a tumour promoter [43,44]
	IIIc	EMT. Found in mesenchymal tissues. Tumourigenic [43]
	FGFRα	Contains autoinhibitory IgI domain which results in a lower affinity for FGFs and decreased signalling compared to FGFRβ [45,46]
	FGFRβ	Higher affinity for FGFs and enhanced signalling. Increases proliferation and linked to tumourigenesis [43,45,46]
	Soluble receptors	Can be found in locations in the cell other than the cell membrane. Precise function unknown [43,45]
	C1, C2 and C3	C3 has the most transforming activity, C2 has moderate transforming activity and C1 has the least transforming activity. C3 implicated in oncogenesis [47]
	Deletion of the VT motif	Prevention of binding of some effector molecules. Suggested to be unable to activate the downstream Ras/MAPK signalling pathway [48]
*Vasohibin-1*	VASH1A and VASH-1B	Both are anti-angiogenic. VASH-1A promotes normalisation of abnormal tumour blood vessels. VASH-1B prunes vasculature [49,50]
*Vasohibin-2*	355aa	Predominantly expressed in HUVECs. Function unknown [51]
	290aa	Anti-angiogenic activity [52]
	311aa, 156aa, 117aa, 104aa	Function unknown
*HIF-1α*	HIF-1αΔ11	Promotes tumorigenesis through enhancement of HIF activity [52]
	HIF-1αΔ12 and HIF-1αΔ11&12	Inhibits dimerisation of HIF-1α and HIF-1β. Act as dominant regulators of HIF-1 transcription [53]
	HIF-1αΔ14	Less potent activator of HIF-1 transcription than canonical form of HIF-1α [54]
	HIF-1α^417^	Amplifies HIF-1β-mediated transcription of *EPO* gene [53]
	HIF-1_TAG_	Function unknown
	HIF-1α Alt1	Function unknown
*HIF-3α*	IPAS	Dimerises with HIF-α subunits but cannot initiate transcription of HIF target genes, such as *VEGF*. Dampens angiogenesis [53,55,56]
	HIF-3α4	Forms complex with HIF-1α and prevents HIF transcription. Hampers angiogenesis and proliferation [55]
*Ang-1*	1.5 kb, 1.3 kb	Bind to Tie-2 and induce its autophosphorylation [57]
	0.9 kb, 0.7 kb	Bind to Tie-2 but do not induce its autophosphorylation [57]
*Ang-2*	Ang-2B	Not precise function but indication of inactivation of the vasculature and vascular remodelling [58]
	Ang-2_443_	Antagonist of Tie-2 signalling activation. Suggestive role in the regulation of inflammatory processes [59]

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
