# Peer review of "Alternative Splicing in Angiogenesis"

_ijms, 2019, doi:10.3390/ijms20092067_

Round 1

Reviewer 1 Report

The review "Alternative Splicing in Angiogenesis" by Elizabeth Bowler and Sebastian Oltean describes in detail known splice isoforms of angiogenesis-relevant genes and, as far as known, their cellular function. As angiogenesis contributes significantly to the development and complication of multiple pathologies and as our understanding of the importance of alternative splicing events has been rapidly increasing recently, this review will certainly be of interest to a broad readership.

Major comments:

1) The individual splice isoforms are very well described, but especially in genes with multiple splicing events, it is sometimes difficult to follow the review. For a better visualization, it would be good to show splicing schemes for the individual isoforms, preferably with an indication of which protein domains are encoded by which exons.

2) The splicing factors/RNA-binding proteins responsible for the regulation of the respective alternative splicing event are mentioned only in a few sections of the review. In cases where no factors are mentioned, is the regulation completely unknown? If not, please go into more details on known factors. Are there splicing factors that control several angiogenesis-relevant splicing events? Are relevant splicing factors altered in pathologies driven by dysregulated angiogenesis?

3) Angiogenesis-relevant genes have a high potential as drug targets. Please comment on the therapeutic potential of exploiting alternative splicing. Are there therapeutics that specifically target individual splice isoforms or modulate splicing patterns?

Minor comments:

Page 1, Line 33: change ribonucleoparticels to ribonucleoprotein particles

Page 4, Line 116: miRNA regulate are one level of posttranscriptional regulation. Please give more examples instead of “etc.” and provide references.

Page 7, line 22: What does “the splicing events that were originally accepted” mean?

Page 9, line 121: How is the isoform NRP-1d7 produced? Exon skipping or maybe alternative splice site usage?

Page 11, line 216: How does a deletion of 6 nt create a frame shift?

Page 12, line 296: Which exons are skipped in HIF-1a417? Exon 4 and exon 17?

Page 13, line 322: “inhibitory” is missing for “I”PAS

Page13, line 331: Which alternative splicing events give rise to the isoform HIF-3a4?

Page 14,line 371: Please describe of which exons Ang-1 isoforms consist.

Author Response

Major comments:

1) The individual splice isoforms are very well described, but especially in genes with multiple splicing events, it is sometimes difficult to follow the review. For a better visualization, it would be good to show splicing schemes for the individual isoforms, preferably with an indication of which protein domains are encoded by which exons.

Splicing schemes have been shown for the most known splice variants of each gene in Figure 4.

2) The splicing factors/RNA-binding proteins responsible for the regulation of the respective alternative splicing event are mentioned only in a few sections of the review. In cases where no factors are mentioned, is the regulation completely unknown? If not, please go into more details on known factors. Are there splicing factors that control several angiogenesis-relevant splicing events? Are relevant splicing factors altered in pathologies driven by dysregulated angiogenesis?

Added in splicing factors which are known to regulate alternative splicing of genes. Splice factor regulation of all genes has been looked into but not added to genes whereby there were no returns on literature searches.

3) Angiogenesis-relevant genes have a high potential as drug targets. Please comment on the therapeutic potential of exploiting alternative splicing. Are there therapeutics that specifically target individual splice isoforms or modulate splicing patterns?

Section added 4.0.

Minor comments:

Page 1, Line 33: change ribonucleoparticels to ribonucleoprotein particles

Done

Page 4, Line 116: miRNA regulate are one level of posttranscriptional regulation. Please give more examples instead of “etc.” and provide references.

Done

Page 7, line 22: What does “the splicing events that were originally accepted” mean? Altered

Page 9, line 121: How is the isoform NRP-1d7 produced? Exon skipping or maybe alternative splice site usage?

Added

Page 11, line 216: How does a deletion of 6 nt create a frame shift?

Agreed. Omitted ‘frame shift’ Line 302-303

Page 12, line 296: Which exons are skipped in HIF-1a417? Exon 4 and exon 17?

Line 396 ‘The skipping of exon 10’.

Page 13, line 322: “inhibitory” is missing for “I”PAS

Done

Page13, line 331: Which alternative splicing events give rise to the isoform HIF-3a4?  

Added line 421.

Page 14,line 371: Please describe of which exons Ang-1 isoforms consist.

Line 461: the type of splicing that generates each of them is not clear

Reviewer 2 Report

In the current review article, Elizabeth Bowler and Sebastian Oltean have focused on the key genes that are involved in the angiogenesis and described the various splice isoforms generated by alternative splicing. Overall this is a well written review article and the literature survey is adequate. However, the authors have omitted some key papers which if included would improve the readers knowledge.

Major comments:

1.      Since the manuscript focuses on alternative splicing, it is necessary that the use of alternative promoters and alternative polyadenylation also be included in Figure 2. Therefore, the title can be read as Different alternative splicing mechanisms. This is important because sVEGFR1/sFlt1 generation is based on alternative poly(A) sites located in internal introns/exons (Ikeda T et al., Mol Cell Biochem. 2016).

2.      In section 1.2 of the manuscript when introducing SR proteins and hnRNPs, mention the number of known SRSFs (1-10) and hnRNPs in the mammalian system.

3.      In section 3.1 detailing splicing of VEGF-A, the authors need to describe about-

-         anti-angiogenic VEGF-Ax isoform and its regulation by hnRNP A2/B1 (Eswarappa SM et al., Cell 2014)

-         CAPER-α (a transcriptional coactivator) mediates alternative splicing and controls the shift from VEGF189 to VEGF165 isoform (Huang G et al., Cancer 2012).

-         Repression of VEGFA by microRNAs and hnRNP L (Jafarifar F et al., EMBO J. 2011).

4.      In section detailing Vascular Endothelial Growth Factor Receptors (VEGFRs), mention as to how sVEGFR2 and esVEGFR2 are generated and whether they are structurally similar. sVEGFR1/sFlt1 is considered to be a potent anti-angiogenic factor. Therefore, it is important that the authors describe how sVEGFR1 structurally differs (i.e. devoid of its transmembrane and tyrosine kinase domains) as well as the role of splice factors in its generation.

-         Numerous studies have shown the role of hnRNP D in the regulation of sVEGFR1 (Ikeda T et al. 2011, 2016; Fellows A et al., Cytokine 2013).

-         A recent study has described how VEGF165 along with the transcription factor SOX2 and the splicing factor SRSF2 controls sVEGFR1-i13 expression (Abou Faycal C et al., Scientific reports 2019).

5.      In section detailing Fibroblast Growth Factor Receptors, the authors need to mention exons encoding the IgI and IgII domains. The authors have omitted some of the recent studies relating the various splice factors involved in regulation of FGFRs such as-

-         PTBP1 regulates FGFR1β (Ming Zhao et al., Oncotarget 2019).

-         hnRNP M promotes exon IIIc silencing (Hovhannisyan RH et al., JBC 2007)

-         RNA-binding motif 4 (RBM4) and neuronal polypyrimidine tract-binding protein (nPTB) have been shown to affect the splicing of the FGFR2 IIIc (Liang YC et al., Oncotarget 2015).

Minor comments:

1.      Line 61 reads as “and this is for the reason for the anti-angiogenic label that has been applied to VEGF-A165b” should read as “and for this reason the anti-angiogenic label has been applied to VEGF-A165b”.

2.      In line 96 endothelil should be endothelial.

3.      In line 122 describe how the deletion of seven amino acids leads to NRP-1D7 formation.

4.      In line 150, it is mentioned that “There are 4 types of FGFRs named FGFR 1-4, which have the same protein structure….”. The authors need to make sure because till date five FGFRs have been identified. FGFR 1-3 have a very similar structure while FGFR 4 lacks the trans membrane and intracytoplasmic domains where as FGFR-5 only has two Ig-like domains (Lee PL et al., Science 1989).

5.      In line 216 it is mentioned “ the exclusion of the VT motif creates a frameshift”. How can the deletion of 6 nucleotides cause a frameshift. The protein sequence will still be in frame but devoid of 2 amino acids. Please explain.

6.      In line 229 VEGR-2 should be VEGFR-2.

7.      In line 239, replace micro mRNA with microRNA.

8.      In line 306, replace that with than.

9.      In line 309 delete causes.

10.   The numbering system used for each sub-section has to be in a continuous order.

Author Response

Major comments:

1.      Since the manuscript focuses on alternative splicing, it is necessary that the use of alternative promoters and alternative polyadenylation also be included in Figure 2. Therefore, the title can be read as Different alternative splicing mechanisms. This is important because sVEGFR1/sFlt1 generation is based on alternative poly(A) sites located in internal introns/exons (Ikeda T et al., Mol Cell Biochem. 2016).

Added to figure 2.

2.      In section 1.2 of the manuscript when introducing SR proteins and hnRNPs, mention the number of known SRSFs (1-10) and hnRNPs in the mammalian system.

Added Line 58-59

3.      In section 3.1 detailing splicing of VEGF-A, the authors need to describe about-

-         anti-angiogenic VEGF-Ax isoform and its regulation by hnRNP A2/B1 (Eswarappa SM et al., Cell 2014)

This is not an isoform generated from alternative splicing so has been left out.

-         CAPER-α (a transcriptional coactivator) mediates alternative splicing and controls the shift from VEGF189 to VEGF165 isoform (Huang G et al., Cancer 2012).

Added. Lines 24-29.

-         Repression of VEGFA by microRNAs and hnRNP L (Jafarifar F et al., EMBO J. 2011). Added Lines 33-37.

4.      In section detailing Vascular Endothelial Growth Factor Receptors (VEGFRs), mention as to how sVEGFR2 and esVEGFR2 are generated and whether they are structurally similar. sVEGFR1/sFlt1 is considered to be a potent anti-angiogenic factor. Therefore, it is important that the authors describe how sVEGFR1 structurally differs (i.e. devoid of its transmembrane and tyrosine kinase domains) as well as the role of splice factors in its generation.

      Added. Line 91-98.

-         Numerous studies have shown the role of hnRNP D in the regulation of sVEGFR1 (Ikeda T et al. 2011, 2016; Fellows A et al., Cytokine 2013).

Added. Lines 126-142.

-         A recent study has described how VEGF165 along with the transcription factor SOX2 and the splicing factor SRSF2 controls sVEGFR1-i13 expression (Abou Faycal C et al., Scientific reports 2019).

Added. Lines 146-150.

5.      In section detailing Fibroblast Growth Factor Receptors, the authors need to mention exons encoding the IgI and IgII domains. Done The authors have omitted some of the recent studies relating the various splice factors involved in regulation of FGFRs such as-

-         PTBP1 regulates FGFR1β (Ming Zhao et al., Oncotarget 2019).

Added. Lines 269-274.

-         hnRNP M promotes exon IIIc silencing (Hovhannisyan RH et al., JBC 2007)  

Added. Lines 232-333.

-         RNA-binding motif 4 (RBM4) and neuronal polypyrimidine tract-binding protein (nPTB) have been shown to affect the splicing of the FGFR2 IIIc (Liang YC et al., Oncotarget 2015). Added. Lines 233-237.

Minor comments:

1.      Line 61 reads as “and this is for the reason for the anti-angiogenic label that has been applied to VEGF-A165b” should read as “and for this reason the anti-angiogenic label has been applied to VEGF-A165b”.

Done

2.      In line 96 endothelil should be endothelial.

Done

3.      In line 122 describe how the deletion of seven amino acids leads to NRP-1D7 formation. Done. Line 183

4.      In line 150, it is mentioned that “There are 4 types of FGFRs named FGFR 1-4, which have the same protein structure….”. The authors need to make sure because till date five FGFRs have been identified. FGFR 1-3 have a very similar structure while FGFR 4 lacks the trans membrane and intracytoplasmic domains where as FGFR-5 only has two Ig-like domains (Lee PL et al., Science 1989).

Done. Altered in Line 216-218.

5.      In line 216 it is mentioned “ the exclusion of the VT motif creates a frameshift”. How can the deletion of 6 nucleotides cause a frameshift. The protein sequence will still be in frame but devoid of 2 amino acids. Please explain.

Agreed. Omitted ‘frame shift’ Line 302-303

6.      In line 229 VEGR-2 should be VEGFR-2.

Done.

7.      In line 239, replace micro mRNA with microRNA.

Done.

8.      In line 306, replace that with than.

Done.

9.      In line 309 delete causes.

Done.

10.   The numbering system used for each sub-section has to be in a continuous order.

Done.